# The FANC/BRCA Pathway Releases Replication Blockades by Eliminating DNA Interstrand Cross-Links

**DOI:** 10.3390/genes11050585

**Published:** 2020-05-25

**Authors:** Xavier Renaudin, Filippo Rosselli

**Affiliations:** 1CNRS UMR9019, Université Paris-Saclay, Gustave Roussy Institute, 94801 Villejuif, France; 2Equipe labellisée “La Ligue Contre le Cancer”, UMR9019, Gustave Roussy Institute, 94801 Villejuif, France

**Keywords:** FANC/BRCA pathway, interstrand cross-link (ICL), DNA repair, genomic instability

## Abstract

DNA interstrand cross-links (ICLs) represent a major barrier blocking DNA replication fork progression. ICL accumulation results in growth arrest and cell death—particularly in cell populations undergoing high replicative activity, such as cancer and leukemic cells. For this reason, agents able to induce DNA ICLs are widely used as chemotherapeutic drugs. However, ICLs are also generated in cells as byproducts of normal metabolic activities. Therefore, every cell must be capable of rescuing lCL-stalled replication forks while maintaining the genetic stability of the daughter cells in order to survive, replicate DNA and segregate chromosomes at mitosis. Inactivation of the Fanconi anemia/breast cancer-associated (FANC/BRCA) pathway by inherited mutations leads to Fanconi anemia (FA), a rare developmental, cancer-predisposing and chromosome-fragility syndrome. FANC/BRCA is the key hub for a complex and wide network of proteins that—upon rescuing ICL-stalled DNA replication forks—allows cell survival. Understanding how cells cope with ICLs is mandatory to ameliorate ICL-based anticancer therapies and provide the molecular basis to prevent or bypass cancer drug resistance. Here, we review our state-of-the-art understanding of the mechanisms involved in ICL resolution during DNA synthesis, with a major focus on how the FANC/BRCA pathway ensures DNA strand opening and prevents genomic instability.

## 1. Introduction

Independent of its size, chemical structure and molecular weight, a DNA interstrand cross-link (ICL) has two direct consequences on the chemistry of the DNA: it deforms its spiral structure and links together both strands of the same double helix [1,2]. Consequently, DNA ICLs impede the opening of the double helix and constitute a physical barrier to the progression of both RNA and DNA polymerases. Because ICLs strongly affect the process of DNA replication, they induce high cellular cytotoxicity, justifying the extensive use of ICL-inducing drugs, including cyclophosphamide, diepoxybutane (DEB), melphalan, cisplatin (cis-diaminedichloroplatine(II), CDDP) mitomycin C (MMC) or photoactivated psoralens, in the treatment of diseases due to cellular hyperproliferation, such as cancer, leukemia and psoriasis. Moreover, ICLs also originate spontaneously as byproducts of normal metabolic cellular processes [3] (and references therein). Thus, to survive, cells are equipped with proteins and pathways that allow ICL removal and replication rescue, which we review here.

Excluding the still poorly understood repair mechanisms of the so-called complex lesions, i.e., clusters of several lesions located close to each other on the two strands of a DNA molecule, Jan Hoeijmakers [4] classified five major independent and clearly separated molecular pathways, each potentially representing subpathways dedicated to the elimination of different classes of individual DNA lesions: base excision repair (BER), nucleotide excision repair (NER), mismatch repair (MMR) and the two branches of double-strand break (DSB) repair, homologous recombination repair (HRR) and non-homologous end joining (NHEJ) pathways.

BER, NER and MMR are dedicated to the repair of lesions or abnormalities altering only one of the two strands of a DNA duplex. Thus, following the elimination of the damaged or incorrectly inserted base(s), the resulting discontinuity in the damaged strand, i.e., caused by either a nick in the sugar–phosphate backbone or a single- or multiple-base gap (associated with an ssDNA stretch on the opposite strand), is filled by a DNA polymerase that “reads” the undamaged strand (Figure 1A). Since the BER, NER and MMR pathways restore the structural and informative content of the damaged strand using the unaffected genetic information stored in the opposite undamaged homologous strand, they are generally considered error-free mechanisms. In other words, they normally enable the survival of a genetically stable cell population [4].

To rescue DSBs, lesions that interrupt the continuity of the DNA molecule, the cell uses two major pathways: NHEJ and HRR (Figure 1B). NHEJ reconstitutes DNA integrity by directly sealing the broken extremities of a two-ended DSB, whereas the HRR mediates the strand repair via a more sophisticated pathway that uses sequence homology and postreplicative DNA synthesis. HRR is also involved in the rescue of stalled replication forks in association with a one-ended DSB or even in the absence of a DSB [5,6], acting as a backup system for the replication process. The replication-associated repair associated with a DSB is defined as break-induced replication (BIR) (Figure 1C). Both NHEJ and HRR are associated with a risk of introducing genetic instability. Indeed, if the extremities of a two-ended DSB are not chemically compatible for direct ligation, NHEJ proteins can create the necessary compatibility by adding or eliminating some nucleotides before sealing the break, inducing mutations and small insertions/deletions (INDELs) at the site of the DSB. Although the HRR is an error-free mechanism per se, repetitive sequences in mammalian DNA create the conditions for errors in the “identification” of the correct sequence to use to form the D-loop and restart DNA synthesis [7]. Therefore, when NHEJ and HRR cohabit, as they are in the S and G2 phases of the cell cycle, their activation is under tight control because overactivation of one or the other can have catastrophic consequences for chromosomal integrity [8,9,10,11,12].

Because an ICL most likely damages both strands of a DNA molecule, similar to a DSB, Hoeijmakers-associated ICL repair with HRR [4].

However, compelling evidence indicates that ICL repair is undertaken by a full-fledged and highly sophisticated network based on proteins and pathways of the BER, NER, MMR and HRR mechanisms and inhibits/prevents the activation of the NHEJ machinery. To harmoniously assemble players from different DNA repair teams, a group of proteins that lack direct DNA repair enzymatic activities was selected through evolution to regulate the spatiotemporal repair of ICLs and protect replication from failure: the FANC/BRCA pathway (Figure 2). The inactivation of the FANC/BRCA pathway leads to Fanconi anemia (FA), a genetically recessive and heterogeneous disorder [13,14] originally described in 1927 by the Swiss pediatrician Guido Fanconi [15,16]. FA is clinically characterized by developmental abnormalities, bone marrow failure and reduced fertility. FANC/BRCA pathway inactivation is associated with acute myeloid leukemia and head and neck, breast, ovarian and pancreatic cancers [17]. The FANC/BRCA pathway-defective cells are highly sensitive to ICL-inducing agents, cell cycle checkpoint abnormalities and chromosome-fragility at both random and common fragile sites [18]. The FANC/BRCA pathway is ubiquitously activated in response to any situation of replications stress, including that caused by DNA lesions [18], hematopoietic stem cell mobilization (awakening from dormancy) [19] or unscheduled oncogene activation [20]. However, the cellular and chromosomal consequences of the loss of function of the FANC/BRCA pathway in the former conditions are less deleterious than are those observed in response to ICL-induced agent exposure.

## 2. Origin of ICL in Cells

Although chemotherapeutic agents have been the primary tool in the study of ICL repair in mammalian cells or in vitro systems, every day, each cell of our body undergoes the formation of as many as 10 spontaneous ICLs as byproducts of cellular metabolism [21].

For example, the beta-oxidation of lipids produces aldehydes such as malondialdehyde (MDA) and 4-hydroxynonenal (4-hNE) that create DNA adducts that account for ICLs in both bacterial and mammalian genomes [22,23,24].

More recently, Patel’s laboratory elegantly demonstrated—using mouse models deficient in both FANC/BRCA repair and aldehyde detoxification pathways—additive effects of these deficiencies. Indeed, this group proposed a two-tier protection system in which an acetaldehyde molecule produced during alcohol metabolism is first detoxified by the aldehyde dehydrogenase (ALDH) enzyme family [25,26]. If this first-tier protection fails, acetaldehyde can create an ICL that is repaired by the FANC/BRCA pathway. This work suggests a metabolic origin of ICLs in cells and helps to explain the bone marrow failure “spontaneously” observed in FA patients [27,28]. Notably, failure in acetaldehyde detoxification, due to the inheritance of a dominant-negative allele (rs671) of the aldehyde-catalyzing enzyme acetaldehyde dehydrogenase 2 (ALDH2) that is present in nearly 50% of the Japanese population, is a recognized aggravating condition in Japanese FA patients [29].

Even though aldehydes have recently become a topic of intense investigation in the field of DNA damage, other metabolites can also generate ICLs, such as nitrous acid, a byproduct of stomach activity or dietary sources [30]. Moreover, by creating an aldehyde residue, abasic sites formed by purine hydrolysis and BER intermediates are also important sites of ICLs [31].

## 3. Biochemistry of the FANC/BRCA Pathway

A schematic representation of the FANC/BRCA pathway is presented in Figure 2. It shows the progressive activation of several proteins and modules.

The first module of the FANC/BRCA pathway, the cargo protein FANCM with its partners, is implicated in the local chromatin aggregation of the second module: the FANC core complex. The FANC core complex comprises three subcomplexes, whose assembly allows the constitution of a ubiquitin E3-ligase machinery that transitorily interacts with the E2 enzyme UBE2 T/FANCT to monoubiquitinate FANCD2 and FANCI [32,33]. Once monoubiquitinated (monoUb), FANCD2 and FANCI are assembled in the chromatin-associated FANCD2/FANCI (ID2) nuclear foci [34] required to mediate several steps involving the fourth heterogeneous module, which includes translesional DNA polymerases, nucleases and HRR-associated proteins. Finally, the deubiquitinase USP1 and its partners, in the fifth module, enable the release of ID2 foci from the chromatin, marking the completion of the process [35].

### 3.1. The First Module: FANCM and Its Partners

FANCM (Mph1 in yeast), originally defined as FA-associated 250 kDa protein (FAAP250) [36] and its several partners, including MHF1, MHF2 [37] and FAAP24 [38], represent the first biochemical module of the pathway. We refer the reader to two excellent recent reviews summarizing the current knowledge on FANCM [39,40]. Briefly, FANCM exerts ATP-dependent branch migration activity [41] that is necessary for MMC resistance, but dispensable for the monoUb FANCD2 and FANCI [42], and DNA translocase activity essential for generating monoUb FANCD2 [36]. FANCM acts as a platform that anchors the FANC core complex via an MM1 motif (amino acids (aa) 826–967) and Bloom helicase (BLM) and its associated partners RMI1, RM2, Topoisomerase IIIA (TOP3A) (the BTR complex) via an MM2 motif (aa 1218–1251) [43,44]. The BTR complex, also named the “dissolvasome”, promotes Holliday junction branch migration and the dissolution of recombination intermediates that could lead to harmful sister chromatid exchange (SCE) events [45,46]. FANCM participates in ICL repair and replication rescue in all three scenarios presented below, and it is also an important mediator of stalled fork signaling. Indeed, its activity is induced by ATR-mediated phosphorylation, and in turn, FANCM participates in the optimal activation of ATR [47]. A key function of FANCM is the recruitment of the FANC core complex [43], the E3-ubiquitin ligase that mediates the monoubiquitination of FANCD2 and FANCI, an essential event for ICL repair and replication rescue [36,48,49,50]. Intriguingly, even though FANCM inactivation leads to hypersensitivity to ICL-inducing agents, defective monoubiquitination of FANCD2 and FANCI and chromosome breakage, its loss-of-function is not the bona fide cause of FA [13,51,52]. FANCM mutations are associated with altered gametogenesis and high risk for breast and ovarian carcinomas [51,52,53,54,55,56,57].

### 3.2. The Second Module: The FANC Core Complex

Assembling three subcomplexes, FANCA-FANCG-FAAP20, FANCC-FANCE-FANCF and FANCB-FANCL-FAAP100, the FANC core complex is an E3-ubiquitin ligase that transfers a ubiquitin moiety from the E2 conjugating enzyme UBE2 T/FANCT to FANCD2 and FANCI at lysine 561 and lysine 523, respectively [32,33]. The structure of this complex revealed two FANCL proteins that harbor E3 ubiquitin ligase activity with a RING domain [58].

Mutations in genes encoding proteins of the FANC core complex are found in more than 90% of FA patients [13,14]. In vitro experiments demonstrated that the FANCA-FANCG-FAAP20 subcomplex is dispensable for FANCD2 monoubiquitination [59], and accordingly, several mutations in *FAN*CA do not lead to MMC hypersensitivity [60]. Finally, it has been recently demonstrated that FANCA participates in a single-strand annealing process (SSA, a variant of the canonical recombination process) in a FANC/BRCA pathway-independent manner to back up RAD52 [61], with potential consequences on immunoglobulin formation during the class-switch recombination process in FA [62].

### 3.3. The Third Module: FANCD2 and FANCI Heterodimer

The ID2 heterodimer is the central hub of the pathway. The two proteins seem to be recruited at sites of damaged DNA before their monoubiquitination [63] mediated by the FANC core complex [64,65]. The monoUb ID2 heterodimer then acquires a higher affinity for chromatin [66,67] and forms nuclear foci [34] in a phospho-H2AX-dependent (γH2AX) manner [68] at sites of DNA damage. The ID2 foci act as structural proteins, protecting and stabilizing the forks, as well as a general regulator of the overall process [69,70,71,72]. USP1 mediates FANCD2 and FANCI deubiquitination and allows the release of ID2 foci from chromatin [35], the signal that the ICL has been removed, the DSB sealed and replication completed.

### 3.4. The Fourth Module: The HRR at Work

In addition to protecting stalled replication forks, ID2 foci enable, both directly and indirectly, the recruitment of the fourth heterogeneous module of the pathway, that includes proteins for DNA incision, translesion synthesis (TLS), ICL elimination and BIR, including BRCA2/FANCD1 [73], BRIP1/FANCJ [74,75], PALB2/FANCN [76], RAD51C/FANCO [77], RAD51/FANCR [78], BRCA1/FANCS [79], XRCC2/FANCU [80], XPF/FANCQ [81], SLX4/FANCP [82,83], REV7/FANCV [84] and RFWD3/FANCW [85]. Notably, heterozygous inactivating mutations in most of the genes encoding these proteins increase the risk for breast and ovarian cancers.

### 3.5. The Fifth Module: USP1, the Finisher

USP1, with its partner UAF1, is the deubiquitinase that acts on FANCD2/FANCI and PCNA [35,86,87]. Deubiquitination allows ID2 foci release and the resumption of DNA synthesis by canonical DNA polymerases after their reload onto PCNA upon the completion of translesional DNA synthesis [88].

USP1 loss of function leads to the constitutive monoubiquitination of FANCD2/FANCI, which causes nonspecific and constitutive ID2 binding onto chromatin, leading to an FA phenotype that includes sensitivity to cross-linking agents [89]. This finding reveals that the ubiquitination-deubiquitination cycle is mandatory for a safe response to ICL lesions and replication stress [90].

### 3.6. Other Partners

The FANC/BRCA pathway is embedded in a more complex network involving cross talk among several key DDR proteins whose inactivation is often associated with rare human genetic syndromes featuring developmental abnormalities, elevated cancer risk, DNA damage hypersensitivity and chromosomal fragility.

ATM and ATR, whose loss of function leads to ataxia telangiectasia [91] and Seckel syndrome [92], respectively, are the two master signaling kinases activated in response to DNA DSBs (ATM) or ssDNA stretches (ATR). ATM and ATR, together with the checkpoint kinase and ATR-target CHK1, phosphorylate and activate several proteins of the FANC/BRCA pathway, including FANCM, FANCA, FANCE, FANCD2, FANCI and BRCA1/FANCS [67,93,94,95,96,97,98,99,100].

Originally identified in the same pathway as ATM based on its response to ionizing radiation (IR)-induced DSBs, the MRE11/RAD50/NBS1 (MRN) complex has been associated with radiation hypersensitivity syndromes, A-T-like disease (ATLD, *MRE11* mutation) [101] and Nijmegen breakage syndrome (NBS or Nibrin, *NBS1* mutation) [102,103,104]. The MRN complex participates in ATM fixation at DSBs [105], and its individual components are involved in maintaining the association of the two extremities of a two-ended DSB (RAD50), in the early resection of a DSB (MRE11), and in checkpoint-signaling downstream of ATM (NBS1) [106]. On one hand, MRN activity and/or recruitment to damaged DNA is defective in FANC/BRCA pathway-deficient cells, and on the other hand, MRN promotes functions of the FANC/BRCA pathway in R-loop dissolution [107,108,109,110].

Biochemical and functional links also exist between the FANC/BRCA pathway and the BLM helicase, whose inactivation causes Bloom’s syndrome [110,111,112]. Because of its biochemical activity, the BLM helicase is involved with several DNA repair intermediates that accumulate during ICL repair. BLM can mediate fork reversal [113,114], is involved in Holliday junction reversal during the HRR of a two-ended DSB [45] and participates in the rescue of anaphase bridges [115]. BLM interacts with FANCM, supporting the idea that the latter is the cargo protein that drives its passengers in a timely manner to the correct place. Moreover, BLM assembly on anaphase bridges has been reported to be at least partially dependent on the presence of FANCD2 foci on condensed mitotic chromosomes [116].

The acetylase TIP60 is another key protein of DDR and is involved in the selection of NHEJ or HRR as the mechanism to rescue DNA continuity downstream of a DSB. Biochemically, TIP60 acetylates, among several other targets, lysine 16 of histone H4 (H4K16), which becomes accessible when the broken chromatin relaxes [117]. The K16 acetylation “folds” the H4 histone tail, limiting the accessibility to the dimethylated K20, the docking site of 53BP1, which protects the DSB extremity from nucleolytic resection [117]. A biochemical analysis demonstrated that TIP60, among several other partners, interacts with ATM, MRN complex [118,119] and s FANCD2 [120]. TIP60 relocalizes to IR-induced DSBs in an ATM-dependent manner or to ICL-stalled forks in a FANCD2-dependent manner. A default in TIP60 accumulation at DSBs induced either postreplicatively or associated with replication stress has a critical role in channeling NHEJ repair for DSBs, which leads to chromosomal aberrancies [120,121].

Among the proteins of the BER system, the activity of two glycosylases, NEIL1 and NEIL3, was clearly associated with ICL unhooking and excision from DNA in both a FANC/BRCA pathway-dependent and FANC/BRCA pathway-independent manner [122,123,124,125,126].

Finally, recent works indicate that, in response to ICL-inducing agents, the MCM8/MCM9 dimer [127,128] and the SMC5/SMC6 complex [129,130] are necessary for downstream RAD51 foci assembly and for maintaining a safe chromosome structure in response to both MMC and CDDP. Cells deficient in components of the MCM8/9 or SMC5/6 complexes demonstrate high levels of chromosome rearrangements, including tri- and quadriradials, as well as mitotic and postmitotic abnormalities, i.e., in anaphase bridges and micronuclei, respectively. Although the mechanisms remain unclear, in light of their biochemical activities (helicase for MCM8/9 and chromosomal structural maintenance for SMC5/6), we speculate that these complexes may participate in HRR-associated replication downstream of RAD51-mediated D-loop formation (MCM8/9), thus maintaining the correct assembly of both DSBs and sister chromatids (SMC5/6).

## 4. Evidence for the Essential Role of the FANC/BRCA Pathway in ICL Repair

The description of the cellular, genetic and chromosomal characteristics of FA patients’ cells indicates the central role of the HRR pathway in response to ICL-inducing agents. A comparison of survival, proliferation and cell cycle profiles of primary and immortalized cells from FA patients and healthy donors made clear that the proteins whose loss-of-function was associated with FA were involved in processes activated during the S and G2 phases of the cell cycle to overcome stressful situations. Indeed, FA cells treated with an ICL-inducing agent demonstrated (a) high cellular sensitivity, i.e., reduced proliferation, increased cell death and reduced clonogenicity; (b) an extended time to complete the S phase; and (c) prolonged arrest in G2. Moreover, a cytogenetic analysis revealed that the metaphasic chromosomes from both untreated and ICL-damaged FA cells had more gaps, breaks and complex rearrangements, including radial chromosomes, dicentrics and rings [13,14,17,18]. All previous cellular observations suggest that ICLs stall replication and require the FANC/BRCA pathway to resolve stalling and coordinate the repair of associated DNA breaks promptly and correctly.

A mutagenesis analysis of the *HPRT* locus in FA cells demonstrated that the activity of FANC proteins is required to limit the induction of genomic INDELs, which eventually favor the induction of point mutations in response to ICLs [131]. This observation was later validated with shuttle vectors bearing a single ICL transfected into WT or FA-mutated cells [132]. Importantly, a comet assay, pulsed field gel electrophoresis and γH2AX foci formation analysis validated that, at equimolar drug doses, treatment with ICL-inducing agents is associated with a similar level of DSB induction and repair kinetics in FA and healthy donor cells [110,133,134]. A cytogenetic analysis demonstrated that, whereas non-FA cells recover a normal karyotype, FA cells present with gross chromosomal rearrangements. This means that the intrinsic capability to seal broken extremities together does not fail; however, the choice of both the specific ends to seal together and the mechanism to use to retrieve a DNA structure is altered in the absence of a proficient FANC/BRCA pathway. Recent work using genome-wide sequencing in HAP1 cells showed that long-term deficiency in FANCC was associated with a slight increase in long deletions (>3 bp) and many chromosomal rearrangements, including chromosomal deletions up to 10 kb, inversions and duplications [135].

Successive molecular analyses showed that ICLs activate the master DNA damage signaling kinases ATM and ATR, indicating that ICLs lead to both DSB and ssDNA stretches, whose presence was validated by the observation of phospho-RPA and γH2AX in western blot and immunofluorescence analyses [97]. The presence of ssDNA stretches is the consequence of the stalled replication forks, their reversal and/or the resection of DSBs to create the 3′-overhang necessary for successive RAD51 nucleation, strand invasion and D-loop formation. Downstream of ATR, phosphorylated CHK1 participates in both DNA repair and cell cycle checkpoint activation. In FANC/BRCA-deficient cells, both ATM and ATR/CHK1 have been described as constitutively overactivated in response to ICL-inducing agents [136,137,138], suggesting either a delay in ICL elimination or an accumulation of toxic DNA repair-mediated intermediates. Thus, the prolonged S and G2 phase delay/arrest observed in FA cells seems to reflect the consequence of a normal physiological response. Accordingly, Grover Bagby’s group demonstrated that exposure to equitoxic drug doses resulted in the same G2 delay in both normal and FA cells [139].

Moreover, although not mandatory for their functions, several proteins in the initial part of the FANC/BRCA pathway need to be phosphorylated in an ATM- and/or ATR/CHK1-mediated manner to optimally perform their functions. FANCM, FANCA and FANCD2 were described as the major targets of these signaling kinases [67,140,141,142].

To describe how a cell copes with an ICL to recover the informative content of the DNA and rescue replication, three major—but not mutually exclusive—models have been proposed. The first one, based on several cellular, biochemical and molecular approaches, was proposed in its original version several years ago by Niedernhofer and collaborators [143]. The second is derived from the elegant and sophisticated in vitro reconstitution of the ICL repair steps in a plasmid first demonstrated by Johannes Walter’s group in Boston [144,145]. The last and most recent model, proposed by the groups of Seidman and Wang [146], is based on the use of the DNA-combing technique, which allows the direct monitoring of DNA synthesis [147]. The three main models discussed here are invariably associated with DSB formation and repair via HRR. Alternatively, minor pathways have been proposed or can be inferred by converging the observations showing the ICL repair and replication rescue function that do not depend on DSB formation or repair via HRR. In any case, FANC/BRCA pathway-associated proteins or modules (when the entire pathway is not involved) have master roles in every ICL repair-associated mechanism (Figure 3).

## 5. First ICL Repair Model: Downstream of a Stalled Single Replication Fork

The first model of ICL repair addresses an ongoing replication fork meeting an ICL, stalling and activating ATR/CHK1 and FANCM, which mutually foster and optimize each other’s action. ATR/CHK1 and FANCM mediate the local assembly and full activation of the FANC core complex that, together with the E2-ubiquitin ligase UBE2 T [148], mediates FANCD2 and FANCI monoubiquitination. Interestingly, evidence shows that the MutS complexes MSH2/MSH6 and MSH2/MSH3, known to be involved in mismatch and INDEL loop recognition by the MMR pathway, respectively, can also function as ICL sensors in lieu of FANCM. This FANCM backup system can also activate the ATR-CHK1-FANC core-FANCD2 cascade, supporting the partial monoUb FANCD2 observed in FANCM-deficient cells [149].

In other words, at an ICL-stalled replication fork, we can observe extremely sophisticated and largely bidirectional biochemical and molecular exchanges between MutS, ATR-CHK1 and the FANCM-FANC core complex as they fully and optimally monoubiquitinate FANCD2 and FANCI. ID2 foci on chromatin allow fork protection and the setup of the following steps: the first is the introduction of a nick in the sugar–phosphate backbone in one of the two DNA strands by a nuclease, possibly SLX4/SLX1, MUS81/EME1, XPF/ERCC1 or FAN1 [83,150,151,152,153,154]. This nick creates a one-ended DSB (Figure 3A), whose presence can activate ATM-dependent signaling.

At this stage, after DSB induction, three DNA lesions are apparent (Figure 3A′): (1) the ICL is still at its original location with (2) an associated ssDNA stretch and (3) a one-ended DSB. Consequently, this situation requires (a) the elimination of the ICL from the strand opposite the DSB, followed by the reconstitution of the double helix sequence and structure, permitting successive BIR; (b) the maintenance of the DSB in place; and (c) strand resection to create the 3′-overhang substrate used for BIR. The ID2 foci are the central hub for the spatiotemporal coordination of these events.

On the chromatid opposite the DSB, the DNA duplex harbors a short ssDNA stretch and the ICL. The ID2 complex appears to be necessary to optimally drive nucleases to incise the sugar–phosphate backbone on the opposite side of the ICL relative to the ssDNA region. This step creates a 3′ extremity to reinitiate DNA synthesis and unhooks the ICL, which remains attached to the DNA backbone by only one extremity, with the other one being attached to a short ssDNA sequence. Replication on this chromatid restarts from the 3′ extremity, but it is rapidly stopped by the presence of the unhooked ICL on the coding sequence. However, since the lesion is only present on one strand, the cell can use a translesional polymerase to replicate the sequence past the damaged base [145,155]. This allows the replication to progress at the risk of the possible introduction of incorrect bases opposite the damage, which is the point of origin for the mutations observed in normal cells exposed to ICL-inducing agents (Figure 3A′′).

The switch from a canonical DNA polymerase to a translesional DNA polymerase is associated with the monoubiquitination of a sliding clamp consisting of PCNA [88]. PCNA deubiquitination signals the switch back to canonical DNA polymerases. As a consequence of the replication fork stalling, FANCD2/FANCI and PCNA are monoubiquitinated by the FANC core complex [156] and RAD18/RAD6 [157], respectively, whereas their deubiquitination is mediated by the same deubiquitinase, USP1. In both cases, deubiquitination indicates that the process of DNA repair or the bypass of the lesion was successfully completed. Thus, it is tempting to speculate that USP1 represents the functional link between repair of the one-ended DSB and completion of the replication on the first strand. Notably, USP1-mediated ID2 deubiquitination favors CHK1 dephosphorylation and G2 checkpoint deactivation [95], thereby switching off ICL-associated signaling.

When the double helix bearing the lesion is reconstituted by the filling of the gap, either NER nucleases (XPF-ERCC1 and XPG) or BER glycosylases (NEIL1 and NEIL3) release the unhooked ICL still attached to the short ssDNA, and the resultant nick or ssDNA gap is filled, fixing the mutation [48,125,158]. The reconstituted double helix can now be used as a substrate for the BIR step that will resolve the resected one-ended DSB and rescue the replication (Figure 1C and Figure 3A′′′).

The first step to rescue the one-ended DSB created at a stalled fork is to maintain its association with the site of the original lesion. The activated FANC core-ID2 participates in the optimal assembly of the MRN complex in nuclear foci [110]. The MRN complex is known for its involvement in DSB repair, in which it acts (a) at a structural level to maintain the DSB extremities in front of each other and to properly align the sister chromatid in conjunction with RAD50, the “Velcro” component of the complex [159]; (b) at a biochemical level by initiating the end resection with MRE11; and (c) at a signaling level, participating in the checkpoint activation via NBS1 [96,107,160].

To repair a two-ended DSB, the cell can invoke either the NHEJ or HRR pathway (Figure 1B) [12]. Several factors determine pathway choice, including cell cycle phase (in G1, only the NHEJ is effective), histone posttranslational modifications around the lesion and competition between the 53BP1 and BRCA1 proteins, which are the drivers of NHEJ and HRR, respectively. Indeed, 53BP1 aggregates on DSBs, protecting their extremities against nuclease-mediated resection, whereas BRCA1, driving nucleases such as CtIP, which extends the MRE11 resection, allowing DSB 3′-overhang formation [161]. However, during ICL repair, when a replication fork stalls and collapses, only a one-ended DSB is introduced. It is paramount here that the rescue is mediated by HRR to avoid the deletions and chromosomal rearrangement generated by the NHEJ-mediated sealing of independent one-ended DSBs located either on different regions of the same DNA molecule or in different chromosomes. The FANC/BRCA pathway drives the repair by HRR via FANCD2, which participates in the localization of the acetylase TIP60 to the broken chromatin. In turn, TIP60 acetylates lysine 16 on histone H4 (H4K16), one of the events that restrains 53BP1 accumulation at histone H4K20me, allowing DSB end-resection and 3′ overhang formation [117,120,121]. In the absence of the FANC core complex and ID2 foci, 53BP1 foci aggregate in cells exposed to ICL-inducing agents, colocalizing with phospho-H2AX (γH2AX) foci. As a consequence, one-ended DSBs are “sealed” by NHEJ and not rescued by BIR, leading to gross chromosomal aberrations [117,120,121,162,163]. To finalize the repair of the resected 3′-overhang, RPA molecules cover the overhang and are successively displaced in a BRCA2-dependent manner by RAD51, which will mediate invasion of the opposite strand and D-loop formation, a prerequisite for replication restart.

Importantly, when replication is not completely rescued, ID2 foci remain on chromatin until the end of successive mitosis. Notably, these foci mark incompletely replicated regions that progress beyond S in G2 and separate in twin spots located on the two sister chromatids that segregate at anaphase-telophase at the extremities of DNA bridges. These bridges link the still unseparated daughter cells, outlining their role in protecting the DNA structure at stalled/collapsed replication forks [116,164,165,166,167,168].

In summary, the key points of this first model are arrest and collapse of one replication fork, induction of a one-ended DSB, ICL unhooking, replication traverse and NER/BER-mediated elimination of the unhooked ICL and HRR-mediated (BIR) rescue of replication (Figure 3A).

## 6. Second ICL Repair Model: Converging Forks at Work

The second model is based on the elegant analysis of repair of a single ICL inserted in a plasmid and mediated by whole protein extracts isolated from *Xenopus* eggs [144,145,169,170]. As presented in the seminal work published by Räschle and collaborators [145], the main lines of ICL repair previously described and the role of the FANC/BRCA pathway was validated in this in vitro system. In particular, the work of Walter’s group demonstrated that the different modules of the FANC/BRCA pathway work inside the converging fork pathway in a manner similar to that of the previously described stalled fork-mediated pathway. Nevertheless, some differences characterize the pathways, mainly at the beginning and end of this alternative mechanism (Figure 3B). The repair and duplication of an ICL-damaged plasmid required that two converging forks starting at a single replication origin in the plasmid stall on each side of the roadblock. Since the ICL was introduced asymmetrically in relation to the origin, the authors observed that a first fork approaches the lesion before the other, halting its progression 20–40 nucleotides from the ICL and awaiting the arrival and halting of the second fork on the opposite side of the ICL. The initial stalling of each converging fork was likely caused by the accumulation of positive supercoils ahead of the ongoing CMG helicase complex that unwinds the DNA to allow the progression of the replication machinery. DNA supercoiling is the consequence of an ICL t preventing the opening of the double helix. Taking into account the size of the plasmid (5.6 kb) used in their in vitro setting, the second fork arrives shortly after the stalling of the first one. After the stalling of the second fork 20–40 nucleotides from the ICL, the first one restarts and definitively arrests one nucleotide before the lesion.

The restarted fork progresses to within one nucleotide from the lesion, suggesting that the CMG complex preceding the replication machinery has been removed, an event that probably contributes to fork collapse, allowing the nuclease-mediated introduction of the nick that creates the “expected” one-ended DSB. At this point, a second nick creates a second one-ended DSB on the other side of the ICL on the same DNA strand where the first was introduced (Figure 3B′). Thus, we now have a partially duplicated plasmid with an ssDNA region of approximately 30–40 nucleotides containing the unhooked ICL and a second “opened” plasmid lacking 30–40 nucleotides and presenting two separated one-ended DSBs (resembling a canonical two-ended DSB). These two DSBs will be reconstituted by HRR one at a time as previously described (Figure 3A′′,B′′′).

Thus, the second point made by this in vitro approach is that two DSBs are created and are the necessary substrates to recover two “normal” plasmids. Can we extend these observations into the “real life” of linear chromosomal DNA assembled in a chromatin structure and compacted inside the cell nucleus? Although converging forks are necessary on a plasmid [171], it is difficult to imagine that this case holds true for nuclear DNA. Drugs create ICLs in a sequence-specific manner, such as photoactivated psoralen in an AT-rich region and MMC in a CG-rich context. Therefore, it is reasonable that clusters of ICLs are the rule rather than the exception. How can a healthy cell repair clusters of ICL-damaged DNA without enduring high INDEL rates?

Additionally, Knipscheer’s and Patel’s laboratories [172] demonstrated, using vitro systems, an alternative pathway to remove the ICLs generated by acetaldehyde without the formation of two DSBs. Indeed, they showed that the ICL itself is cut by an unknown process, which may simply be induced by the mechanical force generated during DNA unwinding, while DNA replication is performed by the specialized polymerase REV1. Interestingly, the authors showed that the mutation spectrum resulting from the second mechanism was far lower than the spectrum observed for the other repair models [172]. Thus, REV1 appears to be an important player during DSB-independent ICL removal [129,155,173,174,175].

Therefore, it seems reasonable to think that the converging fork model represents an important, but nevertheless minor ICL repair mechanism in mammalian cells. This point of view is supported by data from the Siedman and Wang groups suggesting a third mechanism to resolve ICLs. However, circular DNA is present in multiple copies in mitochondria, and it is evident that mitochondrial DNA is subjected to ICL formation, but how DNA lesions are repaired in this particular organelle remains poorly understood. It is noteworthy that at least part of the FANC/BRCA pathway also localizes inside mitochondria [176,177], and moreover, damaged mitochondria, as well as altered mitochondrial metabolism, are major features of FA cells [178,179,180,181]. It is tempting to speculate that the model summarized here may also apply for the replication of mitochondrial DNA [182].

## 7. Third ICL Repair Model: ICL Traverse

Based on DNA single fiber analysis [147] conducted by spreading and/or combing of DNA extracted from human cells following the induction of ICLs by the photoactivation of biotin-labeled psoralens [183,184,185], the groups of Seidman and Wang suggested that the major pathway involved in replication-associated ICL repair depends on a FANCM-mediated ICL traverse [146]. A single-molecule analysis allowed them to observe that, in the large majority of cases, the fork encountering an ICL was able to “traverse” it without repairing the lesion (Figure 3C). In other words, the replisome was able to reassemble on the other side of the ICL to continue replication, an event largely dependent on the translocase activity of FANCM. According to these observations, in healthy cells, ICL traverse would account for 50–75% of the observed events; a single fork arrest was observed near an ICL in 15–20% of the cases, whereas only approximately 5–15% of the measured events were associated with converging forks [146]. Several known FANCM partners facilitate ICL traverse, notably PCNA and BLM [186,187]. Moreover, to traverse the ICL, the replisome complex undergoes extensive remodeling, which is dependent not only on FANCM and ATR, but also on FANCD2 in a monoubiquitination-independent manner [188,189]. Accordingly, FANCD2 and FANCI are recruited to chromatin before they are monoUb [188,189]. Consequently, the ICL traverse pathway appears to be FANC core complex-independent.

Notably, ATR- or FACM-deficient cells switch to the one-ended DSB mechanism to rescue ICL-stalled replication. Indeed, following ATR or FANCM depletion, DNA single fiber analysis revealed that the frequency of events representing ICL traverse falls to 10–20%, that of ICL-stalled single forks increases to 50–70%, and the frequency of converging fork events remains unaltered [186,187,188,189]. Thus, it seems that FANC core complex activation and FANCD2/FANCI monoubiquitination represent the events leading to the shift from ICL traverse to replication fork collapse, one-ended DSB formation and BIR-mediated replication rescue.

Alternatively, because of its branch migration activity, FANCM allows fork reversal and stabilization, allowing ICL incision from one strand by the BER and/or NER proteins, creating an unhooked triple helix structure that permits the “traversal” of the lesion by the reversed fork (Figure 3D) [41].

## 8. Concluding Remarks and Future Directions

In summary, (Figure 3), ICL repair starts when the progression of an ongoing replication fork is first delayed by the accumulation of positive DNA supercoiling ahead of the replication machinery and is finally arrested when the fork physically encounters the ICL. The stalled fork activates a first response associated with the mutual and synergistic activation of ATR signaling and FANCM, which represents the first hub of the process of ICL elimination that will allow resumption of DNA replication. ATR and FANCM function activation initiates all the downstream choices. In particular, FANCM drives ICL traverse (Figure 3C) and protects the stalled forks, either by constituting the platform for ICL resolution via the formation of a one-ended DSB (Figure 3A), by allowing time for arrival of the second fork from the opposite side of the ICL (Figure 3B) or by stabilizing the reversed fork (Figure 3D).

Downstream of ATR/CHK1 signaling, the activation of the FANC core complex leads to the transfer of ubiquitin from UBE2 T/FANCT to FANCD2/FANCI and the subsequent assembly of monoUb ID2 foci onto chromatin in close proximity to the ICL and DSBs. These foci are a second hub of ICL repair. ID2 foci are responsible for fork protection, DNA nick formation, TLS and the assembly of the HRR machinery, and they signal the end of ICL repair upon their release from chromatin by initiating USP1-mediated deubiquitination. Alternatively, by combining BER/NER/TLS and, eventually, HRR, cells may eliminate ICL and restart replication without using DSBs as mandatory intermediate substrates, even when, as a consequence of fork reversal, structures resembling DSBs form and accumulate (Figure 3D).

It is clear that the choice of the strategy used to resume DNA synthesis after removing the ICL blocking the progress of the replication fork depends on the ability of the cell to choose the best option at each node, as in an algorithmic decision process (Figure 4). The first part of the algorithm drives the choices of how to repair the ICL depending on the presence of one or two forks stalled by the lesion. The second part of the algorithm is dedicated to the repair, if necessary, of the one- or two-ended DSB that form after ICL unhooking. Finally, replication resumes after the ICL is completely unhooked.

Independent of the pathway choice, at the end of the process, the only consequence of an ICL will be a point mutation, if any, on both sister chromatids of a replicated chromosome. Consequently, the two daughter cells will carry a heterozygous mutation on each allele where the mother cell had an ICL. In contrast, in the absence of a functional FANC/BRCA pathway, only one of the two chromatids present a point mutation, whereas the DSB on the other strand is sealed with either a distal DSB on the same chromatid or on another chromatid, leading to the formation of INDELs and gross chromosomal rearrangements, as reported by the Moustacchi group twenty years ago [131,190].

In conclusion, the field of ICL repair has undergone tremendous advances. However, several questions remain, including the following:

What are the elements that determine the choice of the pathway to cope with ICLs? Is it the timing of the replication, the local structure (condensation) of the chromatin, the local sequences in the DNA, the presence of secondary DNA structures (G-quadruplexes, R-loops or hairpins) or the presence of transcriptional machinery?

The FANC core complex monoubiquitinates FANCD2/FANCI only after recruitment to chromatin [63], and this focalization is not necessary for ICL traverse [188,189]. Therefore, is FANC core complex-mediated FANCD2/FANCI monoubiquitination involved in the choice between the traverse mechanism and the other pathways?

Similarly, undetermined are the mechanisms that allow FANC core complex recruitment and activation and the roles of the histone code during ICL repair.

Several works have reported that the inactivation of either the SMC5/SMC6 complex [129,130] or MCM8/MCM9 [127,128] leads to FA-like cellular phenotypes, including cellular and chromosomal hypersensitivity to ICL-inducing agents, as well as cell cycle and mitotic abnormalities. Both SMC5/SMC6 and MCM8/MCM9 appear to be involved in a step downstream of RAD51 foci formation. The nature and specificity of their roles, as well as their eventual interplay in the completion of the ICL repair process, remain to be elucidated.

Last, but not least, a mechanistic question remains unanswered in the converging forks pathway: how does the second delayed fork signals its position to “awake” the first fork stalled at 30–40 nt before the ICL? 

## Figures and Tables

**Figure 1 genes-11-00585-f001:**
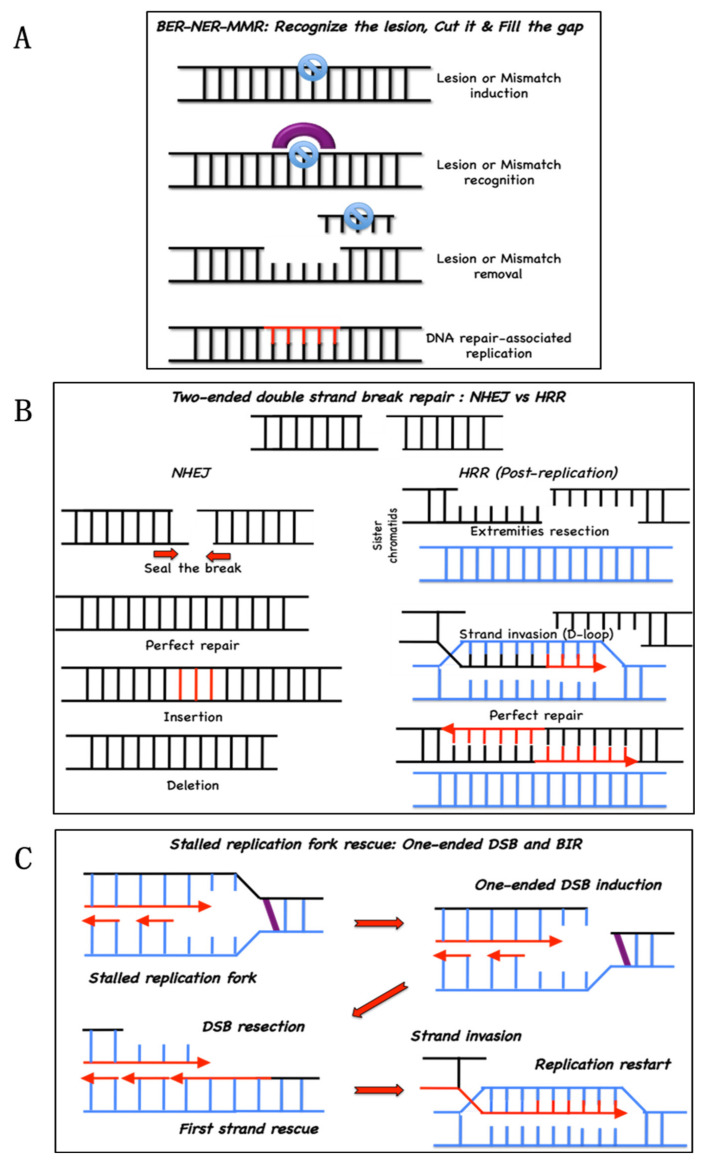
Main DNA repair pathways. (**A**) base excision repair (BER), nucleotide excision repair (NER) and mismatch repair (MMR) repair damaged or mismatched bases. Lesions are recognized and excised, creating a relatively small gap (1–30 nt) that will be filled by a DNA polymerase; (**B**) nonhomologous end-joining (NHEJ, left) or homologous recombination repair (HRR, right) mediate double-strand break repair. NHEJ seals the breaks with compatible extremities. Incompatible extremities can lead to either a perfect repair or to mutations (base insertions or deletions). HRR starts with the resection of the extremities. Resected extremities are used for strand invasion, homology search and D-loop formation in the sister chromatid, allowing perfect repair of the double-strand break; (**C**) HRR safeguards replication. At a stalled replication fork, a single-strand break is generated to allow the subsequent resection and strand invasion in a process named break-induced replication (BIR).

**Figure 2 genes-11-00585-f002:**
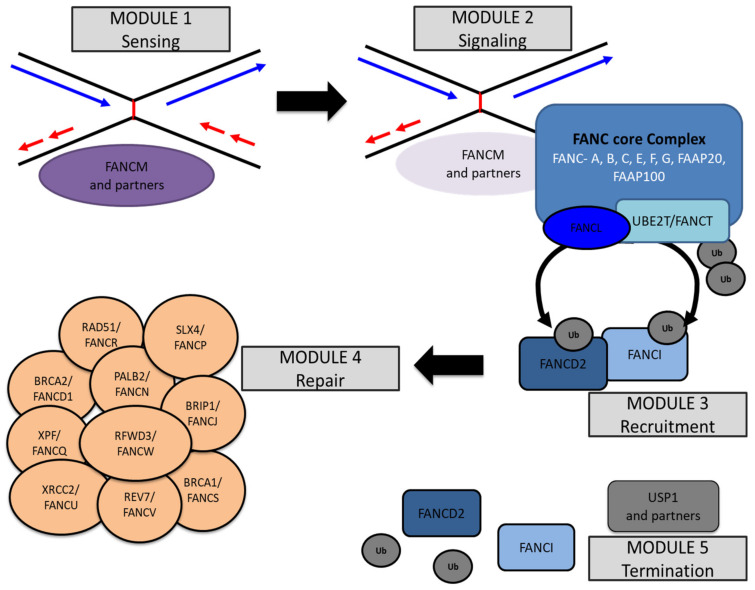
Fanconi anemia/breast cancer-associated (FANC/BRCA) pathway in five modules. Module 1 is composed of the FANCM protein and its partners. FANCM acts as both a translocase and a cargo protein and binds DNA in close proximity to the lesion. Then, module 1 recruits module 2, which is composed of the FANC core complex proteins (names indicated in white) and the E2-ubiquitin ligase UBE2T. The FANC core complex is an E3-ubiquitin ligase, of which FANCL is the catalytic subunit. In response to an DNA interstrand cross-link (ICL), FANCL transfers a ubiquitin moiety from UBE2T onto module 3 proteins (FANCD2 and FANCI). In turn, monoubiquitinated FANCD2 and FANCI, among others, coordinate the recruitment of the proteins of Module 4 to perform ICL removal and restart replication. Once the repair is terminated, module 5 (USP1 and its partners) terminates the reaction by deubiquitinating FANCD2 and FANCI.

**Figure 3 genes-11-00585-f003:**
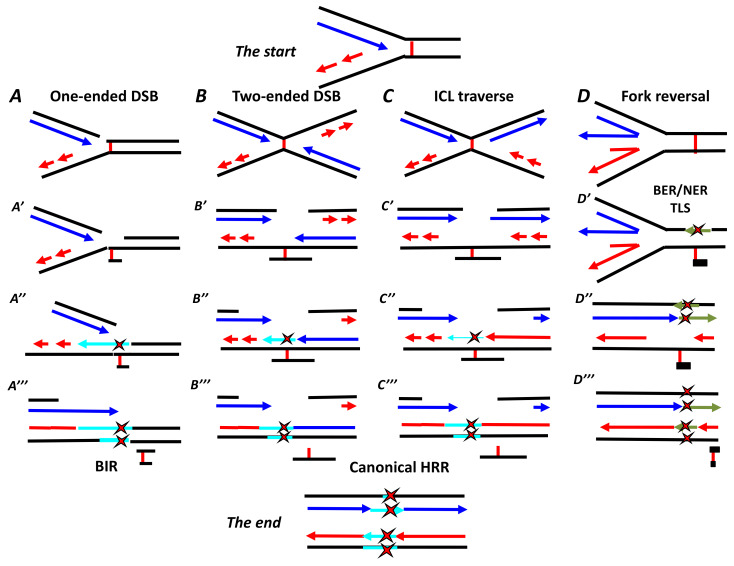
Models of interstrand cross-link (ICL) repair. **(A**) A moving replication fork meets an ICL and stalls. Therefore, by cutting the DNA on one strand, a one-ended double-strand break (DSB) is formed. The ICL is unhooked, forming an ssDNA patch with the ICL on the strand opposite to the DSB (A′). Replication restarts with a possible insertion of an incorrect base in front of the ICL (A′′. After DSB resection, replication resumes from the other sister chromatid using BIR and the repaired chromatid as a template leading, in the end, after the ICL is completely unhooked, to an eventual a mutation on both sister chromatids (A′′′); (**B**) A stalled fork is stabilized until the arrival, from the other side of the ICL, another, convergent fork. Subsequently, at both stalled forks, a one-ended DSB is created, similar to a canonical two-ended DSB, but separated by a 30–40 nucleotide gap that will be rescued by homologous recombination repair (HRR) after the reconstruction of the opposite strand; (**C**) When the replication fork encounters an ICL, the replisome can traverse the lesion without repairing it to continue DNA replication. Later, the ICL will be unhooked (C′), eventually creating, as described above, a one- or two-ended DSB that will be repaired by HRR; (**D**) Alternatively, FANCM can stabilize and reverse the fork(s), allowing time for ICL unhooking before reestablishment of the fork and the “traverse” of the lesion. As in A and B, the repair and replication rescue mediated by pathways C and D lead to the risk of generating and fixing mutations in the duplicated DNA. Red stars indicate the induction of possible mutations by translesional synthesis (TLS) polymerases.

**Figure 4 genes-11-00585-f004:**
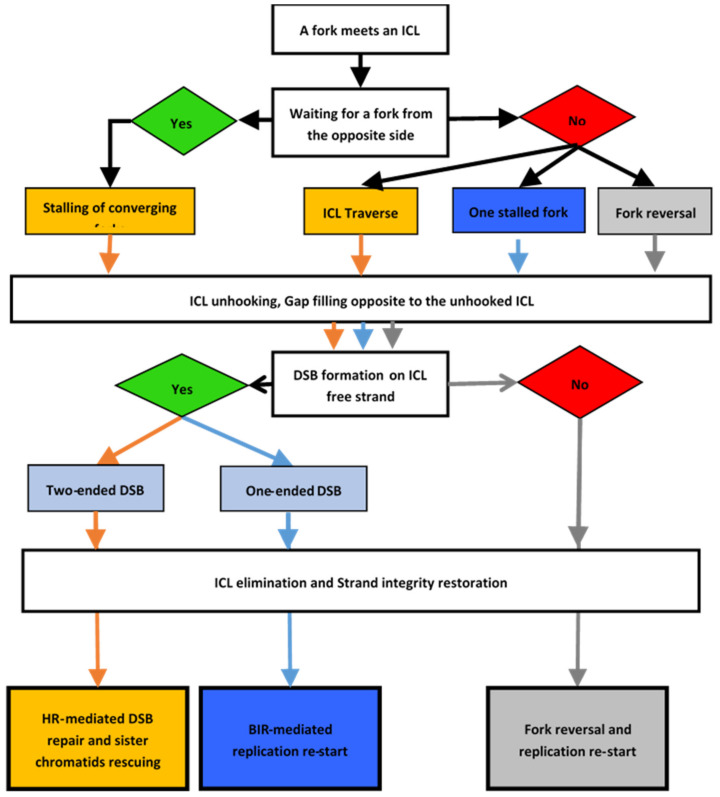
DNA interstrand cross-link (ICL)-repair decision process. Schematic showing the various options offered to a cell when a replication fork faces an ICL. The schematic describes the choices to repair the ICL and the associated DSB until replication is able to resume. The colors illustrate the different paths available to a cell depending on the type of ICL repair.

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
