# Peer review of "The FANC/BRCA Pathway Releases Replication Blockades by Eliminating DNA Interstrand Cross-Links"

_genes, 2020, doi:10.3390/genes11050585_

Round 1

Reviewer 1 Report

The authors summarize the FA pathway using several different models. The title uses several interesting terms including “distancing” and “lockdown”, probably related to the current COVID-19 situation. However, if the authors want to use these terms in the title, they should also describe how the FA pathway uses these concepts to remove and/or repair ICL damage.

Overall, the manuscript has good information regarding how the FA pathway work to remove ICLs. However, the manuscript is rather a brief summary of the FA pathway. Considering that there are many review articles that deal with this pathway, I suggest that the authors include more mechanistic details of this pathway. Additional figures also help readers understand the complicated ICL-repair mechanisms.

Page 2, line 50: The author may remove “mismatches” or describe additional types of abnormalities.

Page 2, lines 55 and 56: I did not understand this sentence very well.

Page 2, line 58: I was not sure what authors meant by “the recombination strategies.

Page 2, line 65: Although explained later, the authors should briefly clarify “intrinsic” and “extrinsic”.

Page 2, lines 74-77: The purpose of this paragraph is not clear to me.

Page 2, line 85: “Linear” may not be appropriate, or please explain. The FA pathway seems to be very complicated and may not be so linear.

Page 5, line 144: “Linear assembling”. Please see above.

Page 5, lines 157-161: The authors may explain how these various proteins work, at least briefly.

Page 5, lines 162-174: I feel that this paragraph is confusing, and it is hard to understand why this paragraph is placed here. I suggest that the authors reorganize or restructure this paragraph, so that readers will see what comes next.

Sections 3, 4, and 5: It is very nice to contrast three different modes of repair employed by the FA pathway. The authors use Figure 3 to explain the differences between those processes. However, Figure 3 is rather simple for such complicated repair processes. I suggest that the authors show additional figures by including various proteins associated with those three different processes. In addition, I feel that the authors should expand these sections to describe the repair mechanisms more in details.

Page 9, line 313: “passage” may be “switch”?

Author Response

Reviewer #1

The authors summarize the FA pathway using several different models. The title uses several interesting terms including “distancing” and “lockdown”, probably related to the current COVID-19 situation. However, if the authors want to use these terms in the title, they should also describe how the FA pathway uses these concepts to remove and/or repair ICL damage.

Overall, the manuscript has good information regarding how the FA pathway work to remove ICLs. However, the manuscript is rather a brief summary of the FA pathway. Considering that there are many review articles that deal with this pathway, I suggest that the authors include more mechanistic details of this pathway. Additional figures also help readers understand the complicated ICL-repair mechanisms.

We thank the reviewer for his positive assessment of our manuscript. We added an additional figure to summarize the “choices” offered to a cell to cope with an ICL. We changed the title and reorganized extensively the manuscript to simplify or complete wherever required.

Page 2, line 50: The author may remove “mismatches” or describe additional types of abnormalities.

We changed the sentence accordingly.

Page 2, lines 55 and 56: I did not understand this sentence very well.

We have now clarified the sentence.

Page 2, line 58: I was not sure what authors meant by “the recombination strategies.

We have now clarified this sentence.

Page 2, line 65: Although explained later, the authors should briefly clarify “intrinsic” and “extrinsic”.

We have rewritten this paragraph for clarification purpose.

Page 2, lines 74-77: The purpose of this paragraph is not clear to me.

We have changed the paragraph accordingly.

Page 2, line 85: “Linear” may not be appropriate, or please explain. The FA pathway seems to be very complicated and may not be so linear.

Page 5, line 144: “Linear assembling”. Please see above.

We remove the term “linear” to clarify. But the FA pathway in per se linear if we consider only the bona fide proteins. i.e In response to a genotoxic stress the Core assembled to monoubiquitinate FANCD2/FANCI which recruit downstream proteins allowing repair. The complexity of the pathway comes from the number of proteins involved.

Page 5, lines 157-161: The authors may explain how these various proteins work, at least briefly.

We add a few details about all these proteins including ATR, ATM, Check1 and the MRN complex as suggested by the reviewer. These details form the new paragraph 3.6.

Page 5, lines 162-174: I feel that this paragraph is confusing, and it is hard to understand why this paragraph is placed here. I suggest that the authors reorganize or restructure this paragraph, so that readers will see what comes next.

 We have changed the paragraph accordingly.

Sections 3, 4, and 5: It is very nice to contrast three different modes of repair employed by the FA pathway. The authors use Figure 3 to explain the differences between those processes. However, Figure 3 is rather simple for such complicated repair processes. I suggest that the authors show additional figures by including various proteins associated with those three different processes. In addition, I feel that the authors should expand these sections to describe the repair mechanisms more in details.

As acknowledged by the referee, our key objective was to offer a "DNA point of view" of the ICL repair instead of a protein point of view that has been largely resumed in several recent published reviews. We want essentially contrast the different modes of the ICL repair and point to the capability of the cell to choice between them.

Page 9, line 313: “passage” may be “switch”?

corrected

Reviewer 2 Report

This review reports on inter-strand crosslinks and the mechanisms that repair them in order to maintain genomic stability. The authors describe the different origins of ICLs, the different mechanisms the cells employ to repair them. While the authors attempt to describe the proteins and pathways that are involved in the repair of ICLs, this review is very difficult to read and understand because of the very dense and convoluted organization of the content. The review needs significant spelling, grammar, and structural edits.

Major Comments

  1. Most sentences in the review text are way too long. It is distracting to decipher the meaning conveyed by the sentence when they are too long. Please construct short and meaningful sentences to help the reader understand the valuable point you are trying to convey. I am only citing one examples here but please not that most of the review has this issue. Example “To harmoniously assemble players from different DNA repair teams, evolution selected a group of proteins lacking direct DNA repair enzymatic activities but having in charge the spatio-temporal regulation of the repair of ICLs and replication rescue: the FANCcore complex and the FANCD2/FANCI heterodimer, which together with several proteins belonging to BER, NER, MMR and mainly HRR constitute the "linear" Fanconi Anemia (FA)/BReast CAncer associated (BRCA) (FANC/BRCA) pathway (Figure 2)” 
  1. In Figure 1B, under NHEJ, the second step mentions “Perfect Repair” which is very misleading. It is essential to change this and clarify that NHEJ is error-prone due to the need for end processing in most cases. In addition, the figure should include an addition step showing this.
  1. Lines 65 and 66. “For both intrinsic (NHEJ) and extrinsic (HRR) reasons, recombination pathways are associated with a risk of 65 introduction of genetic instability.” This statement needs to be explained better as it is misleading.
  1. The term “impede” is not used appropriately in a lot of instances throughout the review and this changes the meaning of the point the authors are trying to make. Please review all sentences that include this word and please use a more appropriate word whenever possible.
  1. Most of the citations used are citing other review articles. Please cite the articles that made the original discovery wherever applicable throughout the review
  1. The sentence in Lines 95-98 doesn’t not make much sense and is limited in describing the FA pathway role in the cell
  1. In line 121, the sub-section called “Origin of ICL in cells” comes out of nowhere. It also ends up being the only sub-section under introduction.
  1. The entire “2. The FANC/BRCA pathway” section is very convoluted. It should be re-written to simply the process to make it easier to follow. The associated Figure 2 is also not clearly organized and is not comprehensive.
  1. Line 162. They explain that there are three models to deal with the ICLs. In the text, the authors just mention the approaches that were taken to discover them. It would be beneficial to mention names or titles for the 3 models. These titles could then be used in the figures and the subsequent sections to simplify things.
  1. Line 315. The authors describe the steps of this repair mechanism, and previously they described the proteins involved. It would be more logical to describe each step with the proteins involved.
  2. The authors also do not spend anytime on identifying gaps and open questions in the field.
  3. Overall, due to the manner in which the review is organized and the convoluted way in which it is written, the concept of ICL repair in terms of the FA pathway is very hard to understand from this review. The authors need to start from the moment an ICL is formed and recognized and review sequentially the steps in the process adding critical updates to the process identified in the last few years. This is not what the reader gets from this review.

Minor Comments

  1. In the title: “Keeping DNA strands distancing” sounds confusing and grammatically wrong. This needs to be modified.
  2. In the abstract, the sentence “Impeding DNA strand opening, interstrand crosslinks (ICLs) represent a major barrier 13 that delays or blocks DNA replication forks progression resulting in growth arrest and cells death, 14 in particular in cell populations undergoing high replicative activity, as cancer and leukemic cells.” is too long. Please break down into 2 sentences for easier understanding.
  3. The entire document is filled typos and major grammatical errors that need to be identified and fixed. I am pointing out a few as examples.
    1. Abstract: “…DNA replication forks progression…growth arrest and cells death”
    2. Line 182. … is stabilized and until the arrival à remove “and”
    3. Line 188. FANCM can stabilize and reverses à can stabilize and reverse or FANCM stabilizes and reverses.
    4. Line 265. “…Suggest that ICLs stall replication and the FAN/BRCA… is necessary…” This is a little confusing, are ICLs also necessary to correctly resolve replication fork stalling and repair?
    5. Line 393. “…single replication origin presents in the plasmid…” present

Author Response

Reviewer #2

This review reports on inter-strand crosslinks and the mechanisms that repair them in order to maintain genomic stability. The authors describe the different origins of ICLs, the different mechanisms the cells employ to repair them. While the authors attempt to describe the proteins and pathways that are involved in the repair of ICLs, this review is very difficult to read and understand because of the very dense and convoluted organization of the content. The review needs significant spelling, grammar, and structural edits.

We thank the reviewer for his valuable input and comments on our manuscript. We agree the manuscript was slightly complicated and we have rewritten it extensively. We hope this new version will provide satisfaction.

We have reorganized the manuscript in the following manner. We separated the endogenous origin of ICL from the introduction in a new section (2). We added an entire sub section (3.6) to describe the role of some key repair proteins including ATR, ATM and MRN complex.

We added a section (4) to separate the evidences of the role of the FANC pathway in ICL repair and to deconvolute the section 5.

Then the three model of ICL repair form now the section 5, 6 and 7 of the manuscript.

A more detailed conclusion is also added including future direction.

Major Comments

  1. Most sentences in the review text are way too long. It is distracting to decipher the meaning conveyed by the sentence when they are too long. Please construct short and meaningful sentences to help the reader understand the valuable point you are trying to convey. I am only citing one examples here but please not that most of the review has this issue. Example “To harmoniously assemble players from different DNA repair teams, evolution selected a group of proteins lacking direct DNA repair enzymatic activities but having in charge the spatio-temporal regulation of the repair of ICLs and replication rescue: the FANCcore complex and the FANCD2/FANCI heterodimer, which together with several proteins belonging to BER, NER, MMR and mainly HRR constitute the "linear" Fanconi Anemia (FA)/BReast CAncer associated (BRCA) (FANC/BRCA) pathway (Figure 2)” 

We thank the reviewer for pointed out the difficulty to read our manuscript. We have corrected extensively the entire manuscript and check English and grammar.

  1. In Figure 1B, under NHEJ, the second step mentions “Perfect Repair” which is very misleading. It is essential to change this and clarify that NHEJ is error-prone due to the need for end processing in most cases. In addition, the figure should include an addition step showing this.

We agree that the NHEJ is considered as error prone even if most of the mutations are due to the alt-EJ. In order to simplify the figure for the reader we correct the figure and indicate the possibility of indels.

  1. Lines 65 and 66. “For both intrinsic (NHEJ) and extrinsic (HRR) reasons, recombination pathways are associated with a risk of 65 introduction of genetic instability.” This statement needs to be explained better as it is misleading.

We have corrected this paragraph for simplification.

  1. The term “impede” is not used appropriately in a lot of instances throughout the review and this changes the meaning of the point the authors are trying to make. Please review all sentences that include this word and please use a more appropriate word whenever possible.

We corrected the use of “impede” through the entire manuscript.

  1. Most of the citations used are citing other review articles. Please cite the articles that made the original discovery wherever applicable throughout the review.

We check that and believe that by having more than 180 citations most of them are not reviews but the original articles. However, we agree that most of the introduction cites reviews but we believe it is appropriate to lead the reader more quickly to the point of the manuscript the FANC pathway and ICL repair.

  1. The sentence in Lines 95-98 doesn’t not make much sense and is limited in describing the FA pathway role in the cell

We corrected the sentence.

  1. In line 121, the sub-section called “Origin of ICL in cells” comes out of nowhere. It also ends up being the only sub-section under introduction.

This section is now a separate section (2) of the manuscript.

  1. The entire “2. The FANC/BRCA pathway” section is very convoluted. It should be re-written to simply the process to make it easier to follow. The associated Figure 2 is also not clearly organized and is not comprehensive.

We deconvoluted this section by rewriting most of the sentences.

  1. Line 162. They explain that there are three models to deal with the ICLs. In the text, the authors just mention the approaches that were taken to discover them. It would be beneficial to mention names or titles for the 3 models. These titles could then be used in the figures and the subsequent sections to simplify things.

We hope that the new version simplifies the reading of the review.

  1. Line 315. The authors describe the steps of this repair mechanism, and previously they described the proteins involved. It would be more logical to describe each step with the proteins involved.

We hope this new version of our manuscript will satisfy the reviewer.

  1. The authors also do not spend anytime on identifying gaps and open questions in the field.

We agree with the reviewer that open questions were missing in the original manuscript and have added these questions in the conclusion (section 8) as future directions.

  1. Overall, due to the manner in which the review is organized and the convoluted way in which it is written, the concept of ICL repair in terms of the FA pathway is very hard to understand from this review. The authors need to start from the moment an ICL is formed and recognized and review sequentially the steps in the process adding critical updates to the process identified in the last few years. This is not what the reader gets from this review.

As stated above we have extensively rewritten the manuscript, and corrected most of the text including misspelling and grammar. We believe this manuscript will provide satisfaction.

Minor Comments

  1. In the title: “Keeping DNA strands distancing” sounds confusing and grammatically wrong. This needs to be modified.
  2. In the abstract, the sentence “Impeding DNA strand opening, interstrand crosslinks (ICLs) represent a major barrier 13 that delays or blocks DNA replication forks progression resulting in growth arrest and cells death, 14 in particular in cell populations undergoing high replicative activity, as cancer and leukemic cells.” is too long. Please break down into 2 sentences for easier understanding.
  3. The entire document is filled typos and major grammatical errors that need to be identified and fixed. I am pointing out a few as examples.
    1. Abstract: “…DNA replication forks progression…growth arrest and cells death”
    2. Line 182. … is stabilized and until the arrival à remove “and”
    3. Line 188. FANCM can stabilize and reverses à can stabilize and reverse or FANCM stabilizes and reverses.
    4. Line 265. “…Suggest that ICLs stall replication and the FAN/BRCA… is necessary…” This is a little confusing, are ICLs also necessary to correctly resolve replication fork stalling and repair?
    5. Line 393. “…single replication origin presents in the plasmid…” present

All of the minor changes have been corrected through the manuscript.

Round 2

Reviewer 1 Report

I feel that the manuscript is improved in this version.

I have several minor comments:

  1. The title does not reflect the contents of the article. The authors should remove “distancing” and “lockdown” from the title, or they include these concepts in the text in a manner that these phrases/concepts were well-integrated in the text.
  2. The addition of Figure 4 is good; however, there is not much explanation for this figure. I feel that the authors should expand this part.
  3. The manuscript still needs major English editing. There are numerous awkward expressions, typos, and grammatical errors.

Author Response

I feel that the manuscript is improved in this version.

We thank the reviewer for his positive comment and for the time is dedicated for the revision of our manuscript. We feel his comments have greatly helped to improve the manuscript.

I have several minor comments:

1. The title does not reflect the contents of the article. The authors should remove “distancing” and “lockdown” from the title, or they include these concepts in the text in a manner that these phrases/concepts were well-integrated in the text.

We have now changed the title to reflect more accurately the content of our manuscript.

2.The addition of Figure 4 is good; however, there is not much explanation for this figure. I feel that the authors should expand this part.

We added an entire paragraph to described more this figure and additional changes have been done in both the figure and legend for clarification purpose.

3. The manuscript still needs major English editing. There are numerous awkward expressions, typos, and grammatical errors.

The manuscript has now been entirely revised by professional service.